# The composition and structure of the ubiquitous hydrocarbon contamination on van der Waals materials

András Pálinkás [1] ✉, György Kálvin [1], Péter Vancsó[1], Konrád Kandrai[1], Márton Szendrő[1], Gergely Németh [2], Miklós Németh[3], Áron Pekker[2], József S. Pap [3], Péter Petrik[1,4], Katalin Kamarás [2], Levente Tapasztó [1] & Péter Nemes-Incze [1] ✉

The behavior of single layer van der Waals (vdW) materials is profoundly influenced by the immediate atomic environment at their surface, a prime example being the myriad of emergent properties in artificial hetero-structures. Equally significant are adsorbates deposited onto their surface from ambient. While vdW interfaces are well understood, our knowledge regarding atmospheric contamination is severely limited. Here we show that the common ambient contamination on the surface of: graphene, graphite, hBN and $MoS_2$ is composed of a self-organized molecular layer, which forms during a few days of ambient exposure. Using low-temperature STM measurements we image the atomic structure of this adlayer and in combination with infrared spectroscopy identify the contaminant molecules as normal alkanes with lengths of 20-26 carbon atoms. Through its ability to self-organize, the alkane layer displaces the manifold other airborne contaminant species, capping the surface of vdW materials and possibly dominating their interaction with the environment.

Layered van der Waals materials have gained a special interest in the last decade because many of them can be exfoliated down to single-unit cell thickness. As the thickness decreases, surface effects become more pronounced; hence, both environmental adsorbates and substrates can substantially influence their properties[1]. In the manufacturing of stacked van der Waals heterostructures, surface contamination may be one of the obstacles, which counteracts the adhesion between the layers and thus limits the achievable configurations[2]. Many experimental investigations and practical uses of vdW materials take place under ambient conditions, making surface contamination inevitable. To this day our knowledge is very limited about the components of such contamination, e.g., the main chemical elements; usually it is only described as 'some hydrocarbon and

adsorbed water'[3]. In recent years it was pointed out that the intrinsic nature of the materials' surface could be totally hindered by contaminants, for example in the case of wetting properties[4,5] or electrochemical activity[6,7].

During the last decade, several groups reported 1D ordered structures on the surface of graphitic materials[8–20], e.g. parallel stripes with a 4–6 nm period, and suggested explanations for the stripe structure as the organized adsorption of molecules. First, Lu et al.[8,9] proposed that the stripes can be built up from a self-assembled layer of molecular nitrogen, later a monolayer of organic chain-like molecules was also suggested[17,18]. Despite the large number of observations[8–20] there is no consensus on the chemical composition, nor the origin of the molecular layer. It is suggested in many of these reports that the

[1]Centre for Energy Research, Institute of Technical Physics and Materials Science, Budapest 1121, Hungary. [2]Wigner Research Centre for Physics, Institute for Solid State Physics and Optics, Budapest 1121, Hungary. [3]Centre for Energy Research, Institute for Energy Security and Environmental Safety, Budapest 1121, Hungary. [4]Department of Electrical and Electronic Engineering, University of Debrecen, Debrecen 4032, Hungary. ✉e-mail: andras.palinkas@ek-cer.hu; nemes.incze.peter@ek-cer.hu

stripe structure possibly follows the zigzag or the armchair direction of the surface's hexagonal lattice. Gallagher et al. directly proved that the stripes are parallel to the armchair direction of the topmost graphene layer[16,17]. Furthermore, it was also reported that, although the stripes can maintain their direction over several micrometers, 60° reorientation of the stripes can occur within the same crystal grains[11,16,19]. Gallagher et al.[16] proposed that a contamination layer could be the cause for the widely reported anisotropic friction on graphene and hBN, but the nature of the molecules forming the ordered surface cover could not be determined. Many groups reported anisotropic friction domains on various vdW materials: monolayer to few-layer graphene[16,17,21–24], hexagonal boron nitride (hBN)[16], molybdenum disulfide ($MoS_2$)[15,24,25], and tungsten disulfide ($WS_2$)[26]. In all these reports, the friction anisotropy exhibits common properties, such as a domain structure, C2 rotation symmetry in each domain, 60° angles between the symmetry axes of adjacent domains, despite the differences in the chemical or mechanical properties of the host surface. In some studies, the phenomenon was explained by periodic strain in the upmost layer[21–23], but no direct evidence for such ripple formation was presented so far. As neither the detailed structure and chemical composition of the contamination layer[16,17], nor the presence of periodic strain ripples[21–23] has been evidenced, the origin of such anisotropic frictional domains on vdW materials is still unclear[24,27].

Here, we show by surface-sensitive methods (AFM, low-temperature STM, grazing angle IR, XPS, and ellipsometry) that a self-organized layer of contaminant molecules grows on the clean surface of various vdW-materials exposed to ambient air. Furthermore, we have determined the subfamily of molecules responsible for the contamination layer: saturated, linear hydrocarbons in the form of normal alkanes, with carbon chain lengths from 20 to 26. We were also able to analyze the structure of the contaminant monolayer down to atomic resolution and controllably manipulate it by AFM.

## Results

### Parallel stripes of contaminants on vdW materials

Our Peak Force QNM® (hereinafter PF) AFM imaging of the flat terraces of van der Waals materials very often reveals parallel stripes with a 3.0–5.2 nm period, similar to earlier reports by other groups[8–20]. We have observed these parallel stripes on different surfaces: cleaved bulk graphite exfoliated monolayer to few-layer graphene on hBN or $SiO_2$, and 70–100 nm-thick flakes of different vdW materials (graphite, hBN, or $MoS_2$) supported on $SiO_2$. The stripes are observable in the PF

imaging modes of surface adhesion, deformation, dissipation, and Young's modulus (for a description of these modes see the "Methods" section). In cases where the tip is sharp enough, the stripes are also apparent in the topography image in AFM (see Fig. 1a and Supplementary Fig. 2)[20]. This stripe structure is not observed on freshly prepared samples and appears after a few days of ambient storage. The presence of these stripes is not restricted to our lab environment in Budapest. We have exposed exfoliated hBN and graphite crystals to the ambient air in San Diego (USA), Sardinia (Italy), and Szántód (Hungary). All of them show stripe patterns and friction anisotropy domains.

We observe the parallel stripes in PF topography maps recorded on flat terraces of three different bulk vdW materials (graphite, hBN, $MoS_2$), with the orientation of the stripes showing a domain-like behavior. We measured domains with a diameter from a few tens of nanometers to more than one hundred micrometers, having the same stripe orientation, whereas on the border of the domains the orientation changes by 60°. In Fig. 1a we present examples of these stripes within one such domain. The stripes are able to run through steps between terraces of highly oriented pyrolytic graphite (HOPG) (Supplementary Fig. 1), climb on bubbles formed on a vdW-heterostructure (Supplementary Fig. 2), and cross the border of a graphene/hBN heterostructure (Supplementary Fig. 3) without breaking. Similar stripes were commonly reported[8–20], but their molecular building blocks have not been identified, since their atomic structure could not be resolved. The universality of their period among vdW substrates differing in chemical composition, lattice constant, thickness, or mechanical strain fields (in the case of bubbles) implicates a common, extrinsic origin.

Since the early years of STM measurements[28–30], a whole field emerged from the investigation of self-organizing molecular monolayers on graphite. This broad literature[31–33] aids us in identifying the few possible molecules and ruling out a large variety of functional groups. We were able to achieve atomic resolution in low-temperature (9 K) STM images (Figs. 1c and 3a) on the contaminant molecular layer enabling us to resolve the inner structure of the stripes: the arrangement of the molecules, their precise length, and lattice periodicity. The individual linear molecules lie side-by-side forming the stripes observed by AFM in the direction almost perpendicular to the molecular axis (light blue arrows in Fig. 1). The molecules in adjacent stripes tend to be shifted by a half-molecule width, perpendicular to the molecule axis. The crystal structure of the contaminant layer varies between centered rectangular (cell angle: 90°) and centered oblique

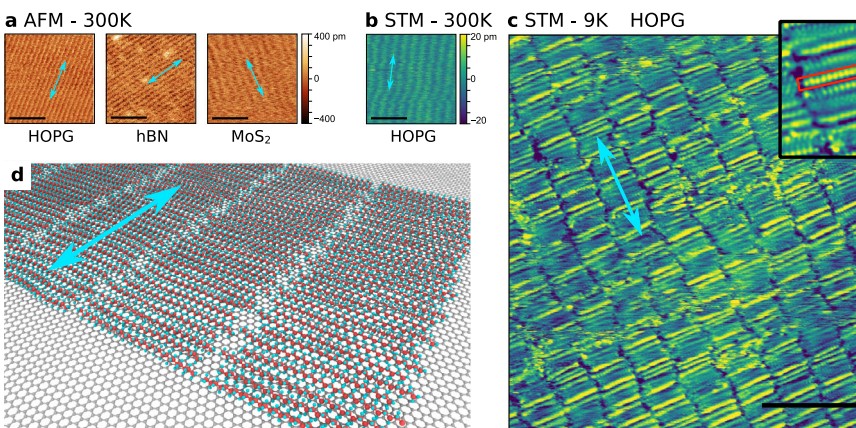

**Fig. 1 | Stripe patterns on vdW materials, due to ambient contamination.**
**a** PeakForce QNM topography maps measured on the surface of bulk HOPG, hBN, and $MoS_2$ crystal terraces showing parallel stripes (Scale bars: 30 nm). **b** Room-temperature STM topography showing the smectic phase of the monolayer (Scale bar: 20 nm). **c** Low-temperature (9 K) STM topography map showing the inner structure of the stripes: the single molecular building blocks and the superlattice along the stripes. (Scale bar: 10 nm). Inset: 6 × 6 nm STM topography image showing the atomic scale structure of a molecular stripe. Single-molecule is shown by the red rectangle. **d** Schematic representation of the linear alkane molecules on a graphene surface. In all images, light blue arrows are parallel to the direction of the stripes, identified by AFM.

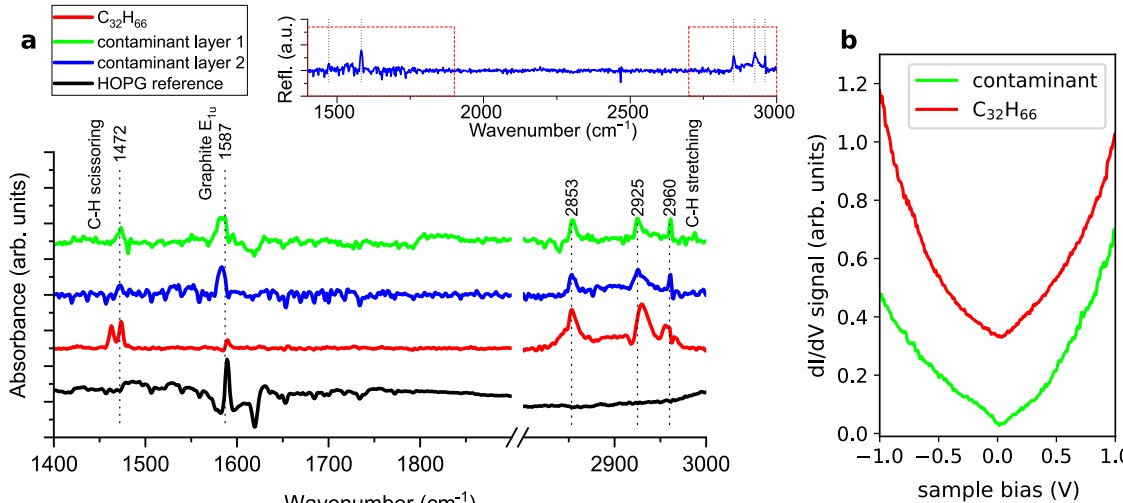

**Fig. 2 | Spectroscopy of the molecular contaminant layer. a** Grazing angle infrared spectra on differently covered graphite substrates, revealing the molecular vibrational modes of the adsorbates. The clean HOPG used as reference (black) shows only the $E_{1u}$ graphite band at 1583 cm$^{-1}$, while the HOPG samples covered with adsorbates (blue, green) or vapor-deposited $C_{32}H_{66}$ layer show the C–H stretching and scissoring modes of alkanes. Inset: The spectrum of "contaminant layer 2" in the full wavenumber range. There are no other peaks aside from the ones presented in the main panel. **b** STM measurement of the tunneling conductance (d$I$/d$V$) on top of the airborne contaminant layer and $C_{32}H_{66}$ (temperature 9 K). The spectra show the typical "V shaped" local density of states signal of graphite, with no sign of molecular states. The spectra are normalized to their respective tunneling conductance used for stabilization. Slight differences in the slope are due to different tips being used on the two samples. Both IR and d$I$/d$V$ spectra are offset for clarity.

(cell angle: 80–85°) regions (see Supplementary Fig. 12). We measured 3.27 ± 0.15 nm for the width of the stripes, which corresponds to an all-trans length of alkyl chains, having 24–25 carbon atoms[34]. This width is in excellent agreement with the stripe width measured by AFM at room temperature (Fig. 1a), where we found 3.4 ± 0.3 nm on graphite and 3.7 ± 1 nm on hBN substrates.

In higher resolution STM images the linear molecules show the atomic periodicity, as bright dots, along their length marked by the red rectangle in the inset of Fig. 1c. Closer inspection of the contrast of adjacent molecules reveals a quasi-periodic modulation of the apparent height along the stripes, with a period of 4–6 molecules. Such periodicity originates from the interaction between the substrate and the molecules, resulting in a 1D superlattice along the stripe. The quasi-periodic nature of the superlattice can be seen over a 35 × 35 nm$^2$ area in Fig. 1c characterized by a period between 1.7 and 2.5 nm. These findings are in agreement with the low-temperature (70–80 K) STM images reported by Endo et al.[35] of an incommensurate crystalline phase of vapor-deposited normal alkanes on graphite. They found that the alkane monolayer is in a smectic phase at room temperature, where only the stripes can be resolved, but the individual molecules cannot, because of their rapid translation parallel to their long axis[35,36]. This is also the case for our own room-temperature STM measurements, where the atomic structure of the molecules cannot be resolved, leading to the image in Fig. 1b.

To image the molecular layer, a large tunneling resistance is needed, above 10 GΩ[28]. In the case of Fig. 1c, we used 20 pA tunneling current and 400 mV sample bias, resulting in 20 GΩ tunneling resistance. Below 5–10 GΩ the tip starts perturbing the molecules and at parameters, which are normally used for graphene measurements (for example 500 mV, 100 pA) the tip penetrates the adlayer, resulting in only a noisy atomic resolution image of graphite[28].

### Spectroscopic measurements

Infrared (IR) spectroscopy can aid in identifying the molecules, by measuring their unique vibrational signature. Furthermore, the presence of a large variety of functional groups, possibly attached to the molecule termini could be also identified. In the search for this signature, we used room-temperature, grazing-angle, and reflectance IR

spectroscopy. After the measurement of one adsorbate-covered substrate, we measured a reference IR spectrum for each sample, by keeping it in the same position but exfoliating the top layers to reveal a pristine surface. In addition, a baseline correction was applied to obtain comparable spectra (see Fig. 2a). Because the large-scale shape and the position of the graphite terraces are not perfectly the same, this results in an observable graphite band ($E_{1u}$ mode−1583 cm$^{-1}$) in all the spectra[37]. This is the only notable feature in the spectra of the freshly cleaved HOPG. In the spectra of contaminant-covered HOPG samples, we could identify the C–H stretching (three bands between 2850 and 2960 cm$^{-1}$) and scissoring modes (weak band at 1472 cm$^{-1}$), whereas other modes typical of functional groups (e.g. –COOH, –OH, halides, etc.) were absent.

The measured IR spectra (blue and green in Fig. 2a) are consistent with normal alkanes as building blocks of the molecular contaminant layer. To further strengthen the case for this, or to disprove it, we have compared the spectra of the contaminant molecules to a monolayer of normal alkanes (dotriacontane, $C_{32}H_{66}$), deposited from the vapor phase onto a clean HOPG surface. In the spectrum of dotriacontane (red spectrum in Fig. 2a) we identified the same peaks as in the contaminant layer. However, the peak of the scissoring mode shows a splitting, which could be due to the presence of thicker layers of dotriacontane, with partial coverage, which could be in a crystalline phase[38]. The contaminant layer shows only single peaks for both the stretching and scissoring modes, suggesting that it is mostly a monolayer. To rule out the presence of carboxylic groups in the airborne layers, we evaporated an arachidic acid ($C_{19}H_{39}COOH$) layer onto HOPG as well. In the IR-spectrum of the artificially deposited carboxylic acid monolayer, we identified the characteristic bands of the COOH headgroup, which are absent in the spectra of the contaminant layers (see Supplementary Fig. 13). Since there are no characteristic bands belonging to common functional groups in IR-spectra of the airborne contaminated layer, we establish that the natural contamination is free of functional groups.

Tunneling conductance (d$I$/d$V$) measurements further support the presence of normal alkanes. This quantity is proportional to the local density of states below the STM tip, at the energy associated with the sample bias. The d$I$/d$V$ curves on both dotriacontane and the

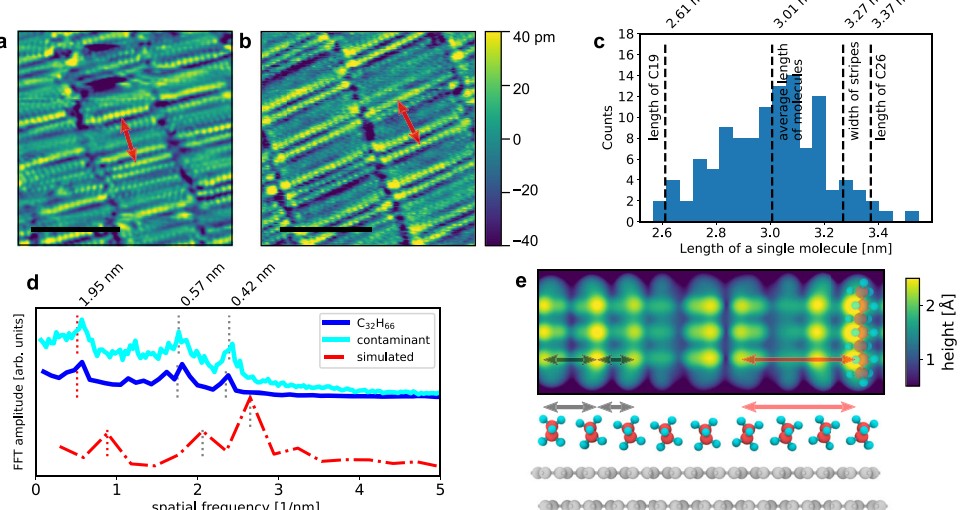

**Fig. 3 | Molecular structure from STM and ab initio density functional theory (DFT) calculations.** STM images of the **a** contaminant and **b** $C_{32}H_{66}$ monolayer on HOPG at 9 K (tip stabilization: 500 mV, 30 pA, scale bars: 4 nm). Red arrows show the quasiperiodic superlattice spacing. **c** Distribution of the individual molecule lengths in the contaminant monolayer. **d** 1D Fourier transforms along the stripes of contaminant (cyan) and $C_{32}H_{66}$ (blue) topographic images showing the periodicities of the molecular spacing and quasiperiodic 1D superlattice. The red curve is the Fourier transform of the simulated STM image **e**), perpendicular to the molecule axis. Dashed lines mark the real space periodicities of the peaks as determined by Gaussian fitting. **e** Top: calculated STM topographic image, at a bias voltage of 500 mV of $C_7H_{16}$ alkanes on a bilayer graphene surface. The calculation cell is 3.41 nm long. Bottom: view along the molecular axis in the calculation cell, showing the equilibrium positions of the alkane chains. The apparent height of the alkanes in STM images is determined by the proximity of the hydrogen atoms to the STM tip and the position of the alkane chains on the graphite surface. Gray lines with arrows show the two interchain distances measurable by STM and in red the characteristic, quasiperiodic superlattice length.

contaminant molecular layer show only the typical charge density of graphite in the energy range of ±1 eV (Fig. 2b). This is expected for normal alkanes since their lowest unoccupied and highest occupied molecular orbitals are more than 7 eV[39] away from the graphite Fermi level. The observation that the d$I$/d$V$ spectra show the "V-shaped" graphite density of states further supports that the molecular layer is only one monolayer thick. Thicker molecular layers, without electronic states near the tip Fermi level, would render STM measurements on top of the molecules difficult because the tip orbital cannot overlap with the $p_z$ orbital of graphite. Stable STM imaging has been demonstrated in cases where there is an insulating monolayer in the tunnel junction, for alkanes[29] and a single layer of MoS$_2$[40]. Alongside the large tunneling resistance required for imaging, the featureless d$I$/d$V$ signal of alkanes is another reason why this remarkable self-organized capping layer has gone unnoticed until now.

## Atomic structure of the contaminant layer

In the following, we show that comparison to $C_{32}H_{66}$ and ab initio modeling of the STM images further strengthens the case for normal alkanes forming the contaminant layer. In Fig. 3 we present higher magnification, low-temperature STM images of the contaminant (Fig. 3a) and the vapor-deposited dotriacontane (Fig. 3b) on graphite. Besides a more ordered arrangement in the artificially deposited, homogenous sample, the strong similarity to the contaminant layer is clear. In the contaminant layer (Fig. 3a), the stripes are formed by long alkyl chains stacked parallel to each other, with no sign of branching, unsaturated carbon-carbon bonds, or benzene rings, similar to the case of dotriacontane (Fig. 3b). Changing the bias voltage between −1 and +1 V does not change the measured topography of the molecules and the relative contrast between neighboring molecules (see Supplementary Fig. 4). This rules out many possible functional groups with higher local density of states and/or higher molecular polarizability[30,41] (e.g. ethers, esters, amines, thiols, carboxylic acids, or –S, –I, –Br chain terminations) which would have brighter contrast. The approximately rectangular structure (angle between the long axis of the molecule and the stripe is 80–85° or 90°, see Supplementary Fig. 12 for the details) in the self-organized layer means that functional groups that interlink by preferential hydrogen bonding[30] (e.g. alcohols or surfactants) can also be discounted, since these prefer a different packing order. In the STM images, alkyl chains with Cl heteroatoms could not be differentiated from normal alkanes[42]. We ruled out that our alkyl chains would be terminated by Cl atoms based on XPS measurements, as the Cl 2$p$ lines near 199 eV were absent from the spectra. We also did not detect any other ubiquitous elements (most notably N, S, Na, etc.), except for traces of oxygen with a concentration below 1% (see Supplementary Fig. 5).

The dotriacontane ($C_{32}H_{66}$) monolayer self-organizes into the same striped structure as the contaminant layer (see also in Supplementary Fig. 12), where the width of the stripe is equal to the length of the molecules, measured to be 4.27 ± 0.16 nm. This is in good agreement with the all-trans length of the molecule[34]. The molecules lie parallel to the HOPG zigzag axis, the direction in which there is no significant contrast variation, whereas along the stripes (HOPG armchair direction) a quasiperiodic 1D superlattice can be observed, similarly to the contaminant layer (red arrows in Fig. 3a, b). The atomic period along a single $C_{32}H_{66}$ molecule is resolved as 16 dots, i.e., the hydrogens of every second carbon atom have the largest apparent height[29,43], but the atomic contrast is smeared periodically every 4–6 molecules because the relative position to the underlying graphite lattice has a strong influence on the local charge density (see Fig. 3e). On the STM image of the contaminant molecules, we could count 10–13 dots with 253 pm (±5%) spacing, in excellent agreement with the distance between alternate carbons in normal alkanes of 251 pm[29]. Compared to dotriacontane, the contaminant layer has a larger variation of the individual molecular length of 2.57–3.55 nm, which corresponds to the all-trans length of normal alkanes with 20–26 carbon atoms. This is in agreement with the width of the stripes (3.27 ± 0.15 nm) measured in Fig. 1c.

In order to quantify the molecular spacing and quasiperiodic superlattice along the stripes, we calculate the Fourier transforms for

each image line parallel to the stripes and average it for the entire image. We plot this 1D Fourier transform of the contaminant and $C_{32}H_{66}$ topographic images in Fig. 3d with cyan and blue lines, respectively. We can identify three peaks at the same positions on both Fourier signals. The peaks at 0.42 and 0.57 nm correspond to molecule–molecule distances. This duality is caused by the different atomic environment of the molecules above the graphite substrate, i.e. when the apparent height maximum is located on the hydrogens on the same side or the opposite sides of adjacent molecules (see black arrows in Fig. 3e). The peak at 1.95 nm corresponds to the quasiperiodic 1D superlattice in the heterogeneous contaminant alkane and dotriacontane layers. The periodicities appearing in the STM images are qualitatively reproduced by the 1D Fourier transform of a simulated STM image of edge-on oriented $C_7H_{16}$ molecules on bilayer graphene (dash-dotted red line in Fig. 3d).

## Simulations of alkanes on graphite

For a deeper understanding of the observed low-temperature ordering of the molecules, we modeled the arrangement of an alkane by using ab initio density functional theory (DFT) calculations implemented in Vienna ab initio simulation package (VASP)[44]. Based on the calculated electronic structures of the different arrangements of the alkanes we also simulated their STM images within the Tersoff–Hamann approximation[45] (see the "Methods" section). Relating to the arrangements of normal alkanes on graphite we shortly discuss two different cases from the literature: (1) the flat-on orientation[29] where the carbon zigzag plane of the alkanes is parallel to the graphite surface and (2) the edge-on orientation[30], where it is perpendicular (Fig. 3e). As we mentioned above, there is a vast literature of STM imaging of self-organized molecular monolayers (and in particular, alkanes) on graphite[29–33], but the overwhelming majority of the experimental papers investigates the phenomenon at room temperature, and at the solvent-graphite interface; therefore, they cannot be directly related to our case. Reports by Diama et al.[36] and Endo et al.[35] revealed temperature-driven transitions between the edge-on and flat-on arrangement of vapor-deposited mid-length alkanes on graphite, which also underline the importance of the delicate balance between molecule–molecule and molecule–substrate interactions. The resolution of this detailed discussion is beyond the scope of the current paper and our DFT computational limit; therefore, we will investigate and discuss solely the edge-on and flat-on cases.

The equilibrium spacing between the adjacent alkane molecules in their 3D solid form does not coincide with the graphite lattice which suggests the presence of a moiré period along the stripes. In our DFT calculations we used the alkane–alkane distance on graphite from previous ab initio calculations by Yang et al.[46], as 0.38 nm for edge-on oriented and as 0.45 nm for flat-on oriented molecules. Based on these values, in our calculations, we modeled 9 (15) molecules in the edge-on (flat-on) orientation on graphite substrate in a 3.408 (6.816) nm long commensurate supercell perpendicular to the molecules. After the geometry relaxation of these alkane–graphite systems, we found that in the case of the edge-on orientation, the molecules have a slight rotation relative to the graphite surface, due to molecule–molecule and molecule–substrate interactions. The calculated STM image of the relaxed geometry is mostly dominated by the hydrogens closest to the STM tip, thus after the rotation, only one side of the molecule's hydrogen atoms is visible on every second C atom (see the rightmost $C_7H_{16}$ molecule on Fig. 3e) in good agreement with the experiments. It is known that besides geometrical effects, electronic effects (originated from the exact position of the alkane chain on the graphite substrate) can also affect the visibility of the molecules on an STM image[43]. This feature appears in our STM simulated images as well, where molecules in the middle of the supercell have similar apparent height as the molecules at the ends of

the supercell even though their hydrogen atoms are lower by 0.2 Å in the z-direction. The combined geometrical and electronic effects result in the variation of the apparent height of the molecules along the stripe, where 2–3 brighter molecules are followed by 1–2 molecules with lower contrast (Fig. 3e). This variability in the simulated STM images can qualitatively explain the experimentally observed 1D quasiperiodic superlattice pattern in both the contaminant alkane layer and the dotriacontane layer.

In order to compare more directly the simulated and measured periodicities, we performed the Fourier analysis of Fig. 3e perpendicular to the alkane chains. Similarly to the Fourier spectra of the measured molecular layers, we can identify three different peaks (periods) in the calculated image (red curve in Fig. 3d): 0.378, 0.485, and 1.13 nm, which are in qualitative agreement with the experimentally measured Fourier components. The first two values correspond to the nearest-neighbor molecule distances, where the furthest hydrogens (0.485 nm) and the alkane–alkane chain separation (0.378 nm) dominate the contrast (gray arrows in Fig. 3e). The third value corresponds to the quasiperiodic modulation of the apparent height in the calculated STM image (red arrow), which is due to the rotation of the alkane chains and electronic effects discussed above. We can conclude that our DFT calculations of alkane chains on a bilayer graphene surface reproduce the main characteristics of the measured STM images. The quantitative differences between the calculated and measured periods are the result of our initial choice of nearest-neighbor molecule distance in our calculations, which define the commensurate supercell. We have also checked that increasing the alkane–alkane distance to 4 Å in our calculation does not change the qualitative agreement with our STM measurement. This enlarged distance also results in a modulation of the apparent height of the molecules, corresponding to the measured quasiperiodic superlattice and the measured two inter-molecule distances of the alkanes (see Supplementary Fig. 11b). It is worth mentioning that the simulated STM image of a flat-on oriented alkane monolayer has qualitatively different features and does not describe our experimental data, in particular, it does not reproduce the observed superlattice (see Supplementary Fig. 11c). However, from our DFT calculations we cannot rule out the presence of mixed phases in the molecule orientation, where the edge-on and flat-on configurations may be intermixed.

## Anisotropic friction domains

Measuring by contact mode AFM the contaminated vdW surfaces reveals a domain structure in the torsion (friction) signal, which cannot be linked to the features in the topography channel and is also present in atomically smooth regions (see Supplementary Figs. 3 and 8b, c). The origin of these frictional domains[15–17,21–26] was not conclusively clarified to date, which hinders the possibility of willingly manipulating or removing them. Our findings establish a direct causal link between the self-organized stripes of the adsorbed alkane layer and the friction anisotropy (Fig. 4a, b). We also demonstrate a precise and reliable method to nano-pattern (switch) the friction domains (Fig. 4c) on the flakes of vdW materials, regardless of their thickness.

In Fig. 4a we present a PF topography map on monolayer graphene supported on hBN. A tripoint boundary with the three distinct orientations (rotated by 60°) of the stripes in the contaminant layer can be observed. The torsion signal of a subsequent contact mode AFM measurement (Fig. 4b) in the same region reveals the three distinct friction domains. The bright spots in Fig. 4a are additional amorphous adsorbates, which are easily swept out by the scanning AFM tip in contact mode. As we have shown above, the stripes parallel to the armchair direction of the surface are formed by linear molecules which lie perpendicular to the stripes, namely the molecules parallel to the zigzag direction. The triplicate nature of the friction domains is caused by the C3 symmetry of the three zigzag directions of the host

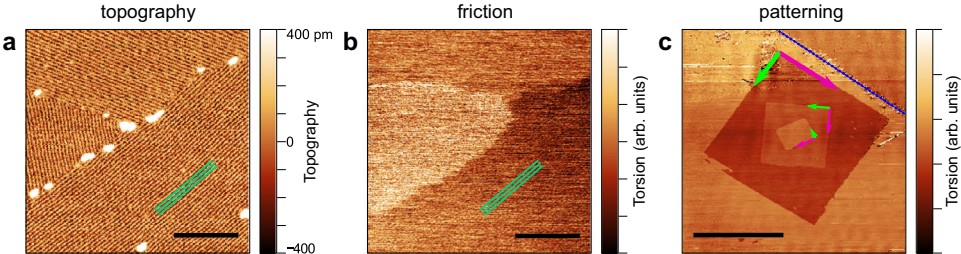

**Fig. 4 | Self-assembled alkane layer as the source of friction domains. a** PF topography image of a tripoint boundary of the stripes on graphene on hBN sample. **b** Contact mode torsion signal on the same area as (**a**), revealing three distinct friction domains corresponding to the three differently oriented molecule domains. (Scale bars on **a** and **b**: 100 nm). **c** Intentionally drawn nested, square-shaped friction domains generated by fast scanning parallel to the three zigzag directions of the substrate (pink arrows). The blue dotted line represents a zigzag-terminated edge of the thick (70 nm) hBN crystal on SiO2, while pink and green arrows show the fast and slow scanning directions of the preceding drawing steps (Scale bar: 10 µm).

surface. The parallel arrangement of the molecules with respect to a given zigzag direction in each domain gives a natural explanation to the commonly observed[16,21,22] easy (zigzag) and hard (armchair) axes of the lateral force anisotropy. We also demonstrate the artificial creation of anisotropic friction domains by the vapor deposition of a straight-chain alkane backboned molecule (alkane or carboxylic acid) monolayer. The dotriacontane ($C_{32}H_{66}$) monolayer on HOPG, presented in Fig. 3b, and arachidic acid ($C_{19}H_{39}COOH$) monolayer on HOPG (Supplementary Fig. 13), display the same domain-like anisotropic frictional characteristics as the naturally contaminated layer (see Supplementary Figs. 7 and 13). By our knowledge, no other reliable, reproducible method was presented so far to controllably create the frictional anisotropy domains.

It has to be noted, that the frictional difference between the domains decreases with increasing applied load (contact force setpoint)[21], thus to get a better contrast, one should scan with low tip–sample force. On the contrary, by scanning with a sufficiently high normal load, we found that the friction domains can be switched by the AFM tip after reaching a threshold, and most importantly, by setting the fast scan direction parallel to the zigzag direction. In Fig. 4c we present the lateral force map on a 70 nm-thick hBN crystal with deliberately patterned friction domains. A zigzag-oriented edge of the crystal (blue dotted line) marks the guideline for the fast scan orientation (indicated by a pink arrow) of the first $15 \times 15\ \mu m^2$ areas, followed by a $7.5 \times 7.5\ \mu m^2$ and a $3 \times 3\ \mu m^2$ area scans where the fast scan direction is changed by 60°(120°). The resulting nested square-shaped friction domains are presented in Fig. 4c. These patterns are stable for up to 6 months. On the other hand, by scanning with very high normal forces, the molecular overlayer can be oriented in any direction, but the formed domain pattern is unstable, it becomes patchy in a few minutes, as the molecules spontaneously rearrange to one of the three preferred zigzag directions of the substrate (see Supplementary Fig. 10 in the case of thick $MoS_2$ and graphene on hBN).

The 60° direction change of alkyl backboned molecules by a scanning AFM tip on graphene was presented by Hong et al.[47]. Similarly to our findings, they stated that the fast scan direction needs to be parallel to the substrate zigzag axis to orient the adsorbed molecules in agreement with the broader literature of nanopatterning of directly deposited alkyl-backboned molecules[47–50]. Relating to the frictional anisotropy domains two previous publications have illustrated methods to switch them, but these findings are limited only to monolayers of graphene[16] or $WS_2$[26]. In the first method, published by Gallagher et al.[16], the final domain orientation on graphene was not predictable, but rather accidental and sometimes unstable. The method of Pang et al.[26] produced the desired and stable patterns, but the interpretation relies on the unproven hypothesis of ripple domains[21]. Here we demonstrated that the changeability of the domain structure is independent of the thickness of the host vdW material, which excludes their strain ripple origin.

## Formation and removal of the alkane layer and friction domains

Despite a large number of experimental observations of naturally occurring friction domains in the literature, as far as we know there is no report where their evolution from the pristine surface is documented. First, we wanted to observe the formation of the contamination layer under ambient conditions. We could only find friction domains and ordered stripe structures on 4–5-day-old samples, stored under lab conditions. We were unable to observe the formation of sub-monolayer self-organized domains or seeds by AFM even during 24 h long measurements. From its fundamental design, AFM is probing the surface of the sample only locally and thus may not be able to fully track the evolution of the adsorbate layer. As a complementary technique, to monitor the layer formation, we used optical ellipsometry which is highly sensitive to the presence of adsorbed species over the whole sample surface. It was reported previously that within 40–60 min an unknown hydrocarbon layer builds up on the freshly cleaved (hydrophilic) surface of HOPG or graphene, which is responsible for the generally observed hydrophobicity of the material[4,5]. However, follow-up studies have shown that after 60 min the thickness of the contaminant layer did not reach a steady state, as evidenced by the further growth of the water contact angle[51]. The growing contact angle due to hydrocarbon contamination was also reported for boron nitride nanotubes[52], $MoS_2$[53,54], and $WS_2$[54]. We have performed similar, but longer-run (up to 2 months) ellipsometry measurements: we found that in spite of the fast build-up of a contaminant layer, the steady state is reached only after 10–14 days (Supplementary Fig. 6).

Based on the combination of AFM and ellipsometry investigations, we propose that in the first 40–60 min an unordered contamination layer grows on the surface (comprising water and diverse hydrocarbons) from higher partial-pressure molecules[55], which on a longer time scale are replaced/interchanged[56,57] by normal alkanes, forming an ordered, stable (lower energy) capping layer. While the atmospheric abundance[58] and vapor pressure of alkanes[59] decrease with their length, the adsorption energy per carbon atom increase with the chain length[57]. This results in the gradual replacement of smaller chain alkanes by longer ones on graphite surfaces[56,57]. Our findings indicate that this results in the growth of the layer comprising alkanes with 20–26 carbon atoms, also due to the limited atmospheric abundance of longer chain alkanes[58]. It is worth noting that similar self-organized stripe structures were also observed to grow on a graphite surface from the liquid phase, particularly when using plastic syringes[18].

Beyond the manipulation of the molecular lattice, and details of the growth dynamics, we show the controlled desorption of the contaminant layer through heat treatment. We annealed the samples at 200 °C for 1 h at $3 \times 10^{-2}$ mbar, which resulted in the disappearance of the friction anisotropy domains (see Supplementary Fig. 9) and the stripe structure. Annealing at 200 °C for 1 h under ambient conditions also leads to the disappearance of domains and stripes but results in

less clean surfaces. This is due to the desorption of the molecular adlayer, as it was shown that monolayers of normal alkanes with 20–26 carbon atoms desorb from graphitic surfaces between 400 and 500 K[60]. The removal of friction domains by annealing is in accordance with the previous findings of Choi et al.[21].

## Discussion

The building blocks of the adsorbate layer predominantly consist of mid-length normal alkanes, consisting of 20–26 carbon atoms. These molecules are very common in the environment: naturally occurring as a mineral (evenkite), present in the cuticle of almost every living organism[61] and they are also a product of crude oil refining. The presence of long-chain alkanes has been shown to be part of the exhaust of internal combustion engines[62] and they are major components of lubricants and mineral oils. The molecules themselves have equilibrium vapor pressures in the order of $10^{-6}$ millibars (decreasing from $6.16 \times 10^{-6}$ to $6.25 \times 10^{-7}$ between $C_{20}H_{42}$ and $C_{26}H_{54}$, respectively). Our results show that they are responsible for the commonly observed stripe structures[8–20] and friction anisotropy domains[15–17,21–26] observed on the surfaces of different vdW-materials.

The observation that normal alkanes are present on graphene, graphite, $MoS_2$, and hBN suggests that these adsorbates could be universally present on vdW materials, at least on those characterized by a hexagonal lattice. Revealing their identity is of key importance for understanding their origin, and their effects on the measured properties. Such crystalline alkane layers may themselves be interesting for some applications[33,63]. One example can be the use of an alkane capping as a barrier, to avoid ambient decomposition of air-sensitive 2D matierals[64], or as gas barriers in conjunction with graphene layers[65].

## Methods

### Sample preparation

Graphite, few-layer graphene, hBN, or $MoS_2$ flakes were exfoliated onto Si/SiO$_2$ substrates with the standard scotch tape method. The freshly exfoliated samples did not show the striped structure, nor friction anisotropy domains. The samples were stored in a standard square plastic box (Ted Pella Inc.) or plastic Petri dish (VWR International, LLC) in ambient laboratory conditions. After 4–5 days of undisturbed storage, the friction domains and the stripes could be observed by AFM. Although not strictly necessary, we found that an hour-long exposure to a $10^{-2}$ Torr vacuum makes the imaging of the stripes easier, possibly because the other volatile contaminations (e.g. water) are removed (see Supplementary Fig. 1). The contaminant layer itself was even stable under UHV ($5 \times 10^{-11}$ Torr, 300 K) for weeks inside the STM chamber.

For the IR, XPS, and ellipsometry measurements we used bulk ($1 \times 1 \times 0.1$ cm) HOPG (Structure Probe, Inc.). The crystals were mechanically cleaved to get a pristine surface. The presence of molecular contamination and the stripe structure was always checked with AFM before IR, STM, and XPS measurements.

Dotriacontane ($C_{32}H_{66}$, Sigma-Aldrich, purity: 97%) layer was deposited onto HOPG substrates from the vapor phase. An appropriate charge of bulk dotriacontane grains was placed in a small glass dish. This small dish, together with the freshly cleaved HOPG was placed inside a larger glass Petri dish and sealed with the glass top. Vapor deposition of the dotriacontane layer occurred upon heating the cell for 30 min to 80 °C, above the melting point of $C_{32}H_{66}$, to reach high vapor pressure. After 30 min the substrate was taken out from the Petri dish and cooled down in ambient air. The glass Petri dish was cleaned with O/Ar plasma etching beforehand, in order to eliminate other organic compounds. Arachidic acid ($C_{19}H_{39}COOH$, Sigma-Aldrich, purity: ≥99%) layer was deposited in a similar method, except that after heating the sealed cell to 80 °C for 30 min, we let the cell cool down to room temperature closed.

### STM

STM measurements were done using an RHK PanScan Freedom microscope, at temperatures of 8.9 and 300 K, in an ultrahigh vacuum at a pressure better than $5 \times 10^{-11}$ Torr. STM tips were prepared by mechanically cutting Pt/Ir (90%/10%) wire and were calibrated on an Au (111) surface. d$I$/d$V$ measurements were recorded using a Lock-in amplifier, with a modulation amplitude of 10 mV at a frequency of 1267 Hz.

### AFM

For the ambient AFM measurements, a Bruker Multimode 8 AFM with different piezo scanners was used. For the precise measurements of the stripe width (calibrated to the graphite lattice, with ±5% relative standard deviation) we used a 9159EVLR scanner with max. ~15 × 15 µm scan area, while for the large scans a 9178JVLR scanner (calibrated to commonly available 1 × 1 µm calibration grid samples) with max. ~150 × 150 µm scan area. Both scanners were used in standard contact, lateral force, and PeakForce QNM modes. In the PeakForce QNM mode, in each image pixel, a complete force curve is measured and the surface adhesion, deformation, dissipation, and Young's modulus measurement channels are extracted. These measurement channels are proportional to the following quantities: the maximum adhesion force between tip and sample, the maximum deformation of the sample surface at peak force, the energy dissipated during a force curve, and a fit to the elastic response of the sample.

We used very sharp (nominal tip radius: 2 nm, spring const: 0.4 N/m) Bruker ScanAsyst-Air AFM tips in PF. For the lateral force/contact mode measurements and for the reshaping of the friction domains we used µMasch HQ:NSC19/Al BS (ntr: 8 nm, sc: 0.5 N/m) and Bruker DNP-10 D (ntr: 20 nm, sc: 0.06 N/m) alongside with the above mentioned ScanAsyst tips. However, the threshold setpoint (loading force) from where we could rearrange the friction domains showed variations and were typical for a given tip-sample combination, but in this manner, the different probe types were not specifically distinguishable.

It is worth noting that in many PF measurements the stripes gave a better contrast in the adhesion or in Young's modulus channels than directly in the topography as it was recognized by Woods et al. as well[20].

We paid special attention to the distance measurement accuracy of our SPMs by calibrating the piezo scanners to the graphite atomic lattice (for the UHV STM and for the 9159EVLR ambient AFM scanners). We always double-check and exclude the possibility of electronic or mechanical noise as the main source of the observed 1D periods by performing measurements on a given area with different scan sizes, scan rates, scan angles, and setpoints.

### DFT calculations

Electronic calculations were performed in the framework of density functional theory (DFT) implemented in the VASP software package[44] using the plane wave basis set and projector augmented wave method[66]. We applied the generalized gradient approximation (GGA) with the parametrization of Perdew–Burke–Ernzerhof (PBE)[67] with added van der Waals (vdW) corrections using the method of Grimme (DFT-D2)[68]. The rectangular supercells consist of the graphite and the alkane molecules (C7) with a period of $a = 12.3$ Å, $b = 34.08$ Å ($a = 14.76$ Å, $b = 68.17$ Å) for the edge-on (flat-on) orientations of the molecules, respectively. The supercells are constructed with a vacuum space of 18 Å along the $z$ direction. Atomic positions were relaxed using the conjugate-gradient method until the forces of the atoms were reduced to 0.01 eV/Å and the cut-off energy for the plane-wave basis set was chosen to be 400 eV. The Brillouin zones were sampled with the Γ-centered K point grid of $6 \times 2 \times 1$ ($6 \times 1 \times 1$) for the edge-on (flat-on) geometries. Simulated STM images were obtained by using the Tersoff–Hamann approximation[45].

## IR

IR spectra were measured by the grazing angle reflectance method at 300 K using a Bruker IFS 66/v vacuum FTIR spectrometer using a Harrick variable angle reflection accessory with 72° angle of incidence and 6 mm aperture. The HOPG samples were exposed to contaminants and measured with up to 6000 scans to increase the signal-to-noise ratio. Reference scans were performed on the same HOPG after exfoliating the contaminated top layer. The reflectance spectrum was calculated by dividing the sample spectrum with the reference measurement. The extensive number of scans and the weak signal (0.05–0.1% change in reflection) necessitated the use of baseline correction. Due to the long-term instabilities of the interferometer, the instrumental artifacts result in broad spectral features[69]. Artifacts are distinguishable from molecular vibrations which appear as narrow peaks in the spectrum. The baseline can be removed by manual baseline correction with proper attention[69,70]. We used a cubic spline curve as a baseline, which we defined by multiple points through the spectra with approximately 50–100 wavenumber step-size to correct the slowly changing background. Then, the absorbance was calculated as $A = 1 - R$ by using the corrected reflection ($R$) spectrum.

## XPS

The surface compositions of the samples were determined in a Kratos XSAM 800 XPS instrument. The samples were analyzed by using an unmonochromatized Al K-alpha source (1486.6 eV) with a take-off angle (relative to the surface) of 90°. Lens-defined selected-area XPS was used to analyze ≈6 mm spot on the sample surface. For the processing of data, the Kratos Vision 2 software provided by the manufacturer was used. The atomic ratio of the elements at the surface was calculated from the integral intensities of the XPS lines using sensitivity factors given by the manufacturer.

## Ellipsometry

$\Psi$ and $\Delta$ ellipsometry spectra have been measured by a Woollam M-2000DI rotating compensator ellipsometer at angles of incidence of 60°, 65° and 70°, where $\Psi$ and $\Delta$ define the complex reflection coefficient $\rho = \tan(\Psi)\exp(i\Delta) = r_p/r_s$ ($r_p$ and $r_s$ denote the reflection coefficients for the polarizations that are parallel and perpendicular, respectively, to the plane of incidence). The measured spectra were evaluated using a two-phase model of a substrate and an overlayer described by the Cauchy dispersion. The optical properties of the substrate were determined on a freshly cleaved sample. The wavelength range was limited to the visible region (450–750 nm) in which the reproducibility of $\Psi$ is better than 0.02, whereas the total ranges of variation are approximately 0.1 and 0.2 for the short- and long-range measurements, respectively.

## Data availability

The raw data of the figures presented in the main text, that support the findings of this study, are available at Figshare, with the DOI identifier: https://doi.org/10.6084/m9.figshare.21294831.

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

## Acknowledgements

The work was conducted within the framework of the Topology in Nanomaterials Lendulet project, *Grant No. LP2017-9/2/2017,* with support from the European H2020 GrapheneCore3 Project No. 881603.

Funding from the National Research, Development, and Innovation Office (NKFIH) in Hungary, through the Grants K-134258, K-131515, FK 125063, FK-142985, Élvonal KKP 138144, TKP2021-NVA-04, TKP2021-EGA-04, and TKP2021-NKTA-05 are acknowledged. P.P. acknowledges support from the H2020 20FUN02 "POLight" grant. We are thankful for the fruitful discussions with Attila Imre and Gábor Piszter regarding the samples from San Diego.

## Author contributions

A.Pa. and G.K. prepared the samples and performed the AFM measurements. A.Pa., G.K., and K.Kan. performed the STM investigation. M.S. and V.P. performed the DFT calculations. G.N., A.Pe., and K.Kam. measured the IR spectra, while M.N. and J.P. performed the XPS measurements. P.P. was responsible for the ellipsometry measurements. L.T. contributed to data analysis and interpretation. P.N.I. conceived and coordinated the project, with assistance from A.Pa. A.Pa. and P.N.I. wrote the manuscript, with input from all authors.

## Funding

## Competing interests

The authors declare no competing interests.
