## [Peer Review File · Nature Communications]

Reviewers' Comments:

Reviewer #1:

Remarks to the Author:

The authors present a study in which they investigate the surface contamination on vdW layered materials. The authors show that a capping layer is formed on graphite, graphene and other 2D materials upon exposure to ambient conditions. The contamination layer is characterized by AFM, low-temperature STM, as well as IR and XP spectroscopy. Combining the results of these techniques the authors conclude that the contamination consists of self-assembled alkanes. This work is thoroughly conducted and well written.

Altogether it is not surprising that ambient exposure alters the surface properties. Moreover, the contamination layer on various vdW materials has been studied before and it has been speculated that the layer is formed by hydrocarbons. (However, as the authors correctly write, some have even speculated the layer to be formed by nitrogen.) Thus, as such the work does not report an unexpected result. Still, it remains very important to remember the presence of these air-borne contaminations. As 2D vdW materials are used extensively, it cannot be stressed too much that the surface properties will inevitably be altered by this capping layer. Most importantly, even so many researchers have speculated that this contamination is formed from hydrocarbons, the precise chemical nature has remained elusive so far. Therefore, the present work provides a very basic and widely important piece of information. I recommend the manuscript for publication after the authors have addressed the following minor issues:

#1 (line 78-80): The authors write that the stripes are observable in [...] adhesion, deformation, dissipation. This seems to be AFM wording. Perhaps the authors could briefly mention that they mean the signal recorded in the given channel? Similar, a PF topography map could be termed differently?

#2 (line 112) The authors write that the molecules in adjacent stripes tend to be shifted by a half-molecule width. Which shift is meant? Perpendicular or parallel to the long axis of the molecule? If parallel is meant, then this is compared with a quantity in perpendicular direction?

#3 (line 217) It is mentioned only in the method's section how the STM calculation is done. It would perhaps be good to mention that briefly in the main text as well?

#4 (line 254) Same as above. A few words more than just 'ab-initio calculation' might be helpful.

#5 (line 299) The minor quantitative differences appear rather major to me. I do not understand the explanation why these distances should differ that much. Why is there a choice in nearest-neighbor molecule distance? Shouldn't this distance be more or less known?

#6 (line 302) The flat-on layer does not describe the experimental data. So? A concluding sentence in the sense of ...this is why we suggest the layer to be formed from edge-on molecules? Why is that? Wouldn't one expect the opposite?

Reviewer #2:

Remarks to the Author:

This paper describes that the identity of the common ambient contamination on the surface of vdW materials such as graphene, graphite, hBN and MoS₂ is composed of a self-organized molecular layer of normal alkanes with lengths of 20-26 carbon atoms, by using the low-temperature STM measurements and infrared spectroscopy. The relation between the contaminant layer and the anisotropic friction measured by AFM is also described. The main conclusion of the paper is well supported by the experimental results. Therefore, I would recommend this paper to be published after a minor revision concerning the following points.

1. Please specify the temperature of the IR measurement, because it is related to the phase of the specimen (smectic or crystalline).

2. As for the IR spectra shown in Figure 2, the splitting of the scissoring mode at 1472 cm⁻¹ is not attributed to the bilayer formation in ref. 40. On the contrary, it seems that the first layer in

contact with the Ag(111) surface shows the split peaks. It may be attributed to the molecule-metal substrate interactions, which may not be the case for the HOPG surface. According to another paper of the same group (J. Phys. Chem. B 2000, 104, 7363-7369), split scissoring mode is attributed to the crystalline phase, not a double layer.

3. The STM contrast of normal alkane layer shown in Figure 3 seems to be well explained by the simulation. However, I wonder that the discrepancy between the distances obtained by the STM observation and DFT calculation in Figure 3 can be serious. The experimentally obtained intermolecular distances 0.42 nm and 0.57 nm are fairly larger than the simulated values of 0.378 nm, 0.485 nm, which is not a minor difference in my opinion. This difference makes the positional relation between the molecule and the substrate atoms different, and therefore, the real molecule-substrate interactions may not be reproduced by the simulation. Moreover, the larger intermolecular distance may allow a more dramatic orientation changes from the edge-on, beyond a slight tilt shown in Figure 3e. Hence, a more detailed simulation or discussion is required concerning this matter, such as the possibility of the further orientation changes. Even if the superlattice is not reproduced for the molecular arrangement with all the molecules with the flat-on orientation as shown in Fig. S11, it does not necessarily exclude the mixture of the edge-on, tilted-on, and flat-on orientation of the molecules.

Reviewer #3:

Remarks to the Author:

In the current paper, the authors have investigated the chemical composition of the airborne hydrocarbon contaminants on 2D materials using grazing angle IR, AFM, STM, ellipsometry and DFT calculation. They conclude that the contaminants are saturated alkane w/ 20-26 carbons that self-organize to form ordered "stripe" on the surface of 2D materials. The characterization and analysis are carefully conducted. The conclusion is supported by the experimental and DFT results. This is a high-quality work on a very important topic. While it has been recognized in the past decade that airborne hydrocarbon contamination significantly impacts the properties of 2D materials, it is still unclear what those contaminants are. Moreover, there has been very limited efforts on this topic. This reviewer suggests that the paper be accepted for publication after the following minor issues are addressed:

1. In grazing angle IR experiments, there should be another control: A hydrocarbon (e.g., SAMs) with functional groups (e.g., COO). This is to make sure the IR has the resolution to detect small amounts of functional groups.
2. While the authors discussed the alkane in the ambient (line 410-418), it is still unclear why it is the alkane with 20-26 carbons. Why not the smaller or larger alkanes?
3. How local environment will impact the conclusion? In other words, how universal is the conclusion? The authors need to discuss on it.

Answers to Reviewers

Reviewer #1 (Remarks to the Author):

The authors present a study in which they investigate the surface contamination on vdW layered materials. The authors show that a capping layer is formed on graphite, graphene and other 2D materials upon exposure to ambient conditions. The contamination layer is characterized by AFM, low-temperature STM, as well as IR and XP spectroscopy. Combining the results of these techniques the authors conclude that the contamination consists of self-assembled alkanes. This work is thoroughly conducted and well written. Altogether it is not surprising that ambient exposure alters the surface properties. Moreover, the contamination layer on various vdW materials has been studied before and it has been speculated that the layer is formed by hydrocarbons. (However, as the authors correctly write, some have even speculated the layer to be formed by nitrogen.) Thus, as such the work does not report an unexpected result. Still, it remains very important to remember the presence of these air-borne contaminations. As 2D vdW materials are used extensively, it cannot be stressed too much that the surface properties will inevitably be altered by this capping layer. Most importantly, even so many researchers have speculated that this contamination is formed from hydrocarbons, the precise chemical nature has remained elusive so far. Therefore, the present work provides a very basic and widely important piece of information. I recommend the manuscript for publication after the authors have addressed the following minor issues:

#1 (line 78-80): The authors write that the stripes are observable in [...] adhesion, deformation, dissipation. This seems to be AFM wording. Perhaps the authors could briefly mention that they mean the signal recorded in the given channel? Similar, a PF topography map could be termed differently?

We thank the reviewer for pointing out that the AFM jargon may be inadequate for a wide readership. The PF topography map can safely be referred to as just topography. We have modified the main text with a brief description of the quantities being measured by the various AFM modes.

Revised main text:

“The stripes are observable in the PF imaging modes of: surface adhesion, deformation, dissipation and Young’s modulus (for a description of these modes see Methods).”

Revised Methods section:

“In the PeakForce QNM mode, in each image pixel a complete force curve is measured and the surface adhesion, deformation, dissipation and Young’s modulus measurement channels are extracted. These measurement channels are proportional to the following quantities: the maximum adhesion force between tip and sample, the maximum deformation of the sample surface at peak force, the energy dissipated during a force curve and a fit to the elastic response of the sample.”

#2 (line 112) The authors write that the molecules in adjacent stripes tend to be shifted by a half-molecule width. Which shift is meant? Perpendicular or parallel to the long axis of

the molecule? If parallel is meant, then this is compared with a quantity in perpendicular direction?

We thank the reviewer for pointing out the need to explain this aspect in more detail. In the text, the perpendicular direction is meant. We have revised the text to make this clearer.

New text in the revised manuscript:

“The molecules in adjacent stripes tend to be shifted by a half-molecule width, perpendicular to the molecule axis. The crystal structure of the contaminant layer varies between centered rectangular (cell angle: 90°) and centered oblique (cell angle: 80-85°) regions (see FigS12).”

Furthermore, we added a new figure to the Supplementary Information, FigS12 regarding the crystal structure of the airborne contaminant and the dotriacontane (C32) monolayers in order to make a clearer statement.

For more details on the crystal structure please read below.

Regarding the homogenous C32 monolayer, it is much easier to pinpoint the exact crystal structure, whereas in the heterogeneous airborne layer we find slightly different arrangements in different regions.

In their bulk form the normal alkanes form two kinds of crystal structures depending on their molecular parity: the even-numbered alkanes exhibit triclinic (cell angle 75-85°), while the odd-numbered alkanes exhibit primitive (or close to the melting point, face-centered) orthorhombic (cell angle: 90°) structure [1]. The molecular parity dependence is more pronounced for short *n*-alkane (<16 carbons) monolayers, as diffraction studies reveal an odd/even alternation such that the even-length *n*-alkanes form herringbone assemblies, while odd-numbered ones form rectangular lamella-molecular backbone structures.

Our particular interest is of mid-length alkane monolayers where the odd/even alternations in morphology are also observed (for monolayers of C16-C21 molecules): the structures are of the centered rectangular (odd) or oblique (even) type [1]. The dependence on molecular-parity is still observable in longer alkanes; however, there is only a 4-5° difference from 90° in the case of even alkanes, which is often disregarded. However, there are reports in the literature for longer, even alkanes (C24 or C32) where the authors stated centered rectangular arrangements [2-3].

As can be seen in FigS12 a-b, in the airborne contaminant layer, thanks to the heterogeneous length of the C20-C26 molecules, we could observe both centered rectangular and centered oblique regions. On the other hand, in the homogenous C32 monolayer we found only the centered oblique structure, where the cell angle was ~84°.

1. Espeau, P., Reynolds, P. A., Bowling, T., Cookson, D. & White, J. W. X-Ray diffraction from layers of *n*-alkanes adsorbed on graphite. *J. Chem. Soc. - Faraday Trans.* **93**, 3201–3208 (1997).
2. Diama, A. *et al.* Structure and phase transitions of monolayers of intermediate-length *n*-alkanes on graphite studied by neutron diffraction and molecular dynamics simulation. *J. Chem. Phys.* **131**, (2009).
3. Endo, O. *et al.* Incommensurate crystalline phase of *n*-alkane monolayers on graphite (0001). *J. Phys. Chem. C* **115**, 5720–5725 (2011).

#3 (line 217) It is mentioned only in the method's section how the STM calculation is done. It would perhaps be good to mention that briefly in the main text as well?

We agree with the Referee, therefore we added further sentences related to STM simulation methods in the revised version of the manuscript. We added the following text to the manuscript:

“we modelled the arrangement of an alkane by using ab initio density functional theory (DFT) calculations implemented in Vienna Ab initio Simulation Package (VASP)⁴³. Based on the calculated electronic structures of the different arrangement of the alkanes we also simulated their STM images within the Tersoff-Hamann approximation⁴⁴ (see Methods).”

#4 (line 254) Same as above. A few words more than just “ab-initio calculation” might be helpful.

We have extended the revised version of the manuscript with a brief description of the calculations. See the new text above.

#5 (line 299) The minor quantitative differences appear rather major to me. I do not understand the explanation why these distances should differ that much. Why is there a choice in nearest-neighbor molecule distance? Shouldn't this distance be more or less known?

We thank the referee for pointing this out. The exact inter-molecule distance is uncertain, as we point out in more detail below. Due to the computationally expensive nature of the problem of reproducing our STM images, we only strive to model the qualitative features of our STM measurements, namely:

- the appearance of brighter and darker molecules and the presence of a quasi-periodic superlattice thereof,
- having two inter-molecule distances in the measurement (see Fig 3)

Our DFT calculations reproduce both of these effects. Our findings are further supported by additional calculations with a larger alkane – alkane distance.

We have removed the word “minor” from the main text, describing the quantitative differences.

Details of the inter-molecule distance:

Despite the great number of observations and the fundamental role of the n-alkane chains in the literature of the self-organized monolayers, the exact value of the molecule-molecule distance is still elusive both experimentally and theoretically. It is related to the fact that the delicate balance between the molecule-molecule, molecule-substrate and molecule-environment forces governs the exact structure of the molecule monolayer. Furthermore, the structure can depend on the temperature and the number of alkane layers as well.

From an experimental point of view, the molecule-molecule distances in 3D, bulk alkane crystals can be exactly known from numerous studies [1]. However, these values cannot be used as such, because the crystal structure of the adsorbed 1-4 layers of alkanes is distinct from the bulk counterpart [2]. The main advantage of the STM investigation is that the molecule-molecule distance of the adsorbed layers on the surface can be measured directly. However, the great majority of the STM examinations of the structure of alkanes on graphite are conducted

on monolayers deposited from solution (from phenyloctane or *n*-decane) [3-4]. The comparison with these experiments is not straightforward in our case, because of the so called “solvent-effect” [5]. The liquids above the deposited solute molecules imply different forces; the formed monolayer may exhibit slightly different packing in different solvents, which means that it is a distinct system to our substrate-vacuum interface [5-6]. It was shown that even in the low temperature STM images, the distribution of the molecule distances can reach the 0.2 nm value on a graphite surface [7].

In our own STM studies, the intermolecular distance is not directly visible, since – as we point out in our manuscript – the largest contribution to the STM image stems from the hydrogen atoms, the apparent height of which is determined by the possible rotation of the alkane chain and its position relative to the graphite below. Thus, a direct measurement of the inter-molecule distance is difficult.

From the point of view of theoretical calculations, similar differences appear in the molecule-molecule distance due to the different calculation methods. For example, classical molecule dynamics calculations (MD) predict 3.8 Å molecule-molecule distance for the edge-on configuration on graphite (Figure 2 in [8]), while DFT calculations define 3.6 Å (Yang et al.[9]). These differences probably arise from the different vdW treatment of the molecule-molecule and molecule-substrate interactions in the MD and DFT simulations. The cited DFT calculations also pointed out that the slope of the interaction energies is rather flat around the energetically most favorable molecule-molecule distance position, which means that there is only a slight energy difference between the structures with different molecular distances. In particular, for the edge-on (flat-on) orientation similar energies appear between 3.5-4 Å (4-4.5 Å) molecule-molecule distances. This wide region of the calculated values can explain the differences in the measured molecule-molecule distances highlighting the importance of the external conditions in the measurements.

As we explained above, defining an exact molecule-molecule distance in general is not straightforward, neither experimentally nor theoretically. In our edge-on geometry we used 3.8 Å, because we found it the energetically best interchain distance in our reference DFT calculations, in good agreement with previous literature data [8]. In this case, the cell with the graphite substrate has 34.08 Å length, containing 9 alkanes (527 atoms together). According to the cited DFT paper [9], we can further increase the molecule-molecule distance in our calculation to approach the experimentally observed value. We have performed new DFT calculations with 4.0 Å molecule-molecule distance in a 68.2 Å supercell consist of 17 alkanes (1159 atoms). We observed qualitatively the same STM image features of the moiré pattern as in the case of the 3.8 Å distance, namely the appearance of brighter and darker molecules and the two inter-molecule distances (see Fig S11) in good agreement with the measurement. Therefore, our new results demonstrate the applicability of our previous conclusions for different molecule-molecule distance as well.

1. Craig, S. R., Hastie, G. P., Roberts, K. J. & Sherwood, J. N. Investigation into the structures of some normal alkanes within the homologous series C₁₃H₂₈ to C₆₀H₁₂₂ using high-resolution synchrotron X-ray powder diffraction. *J. Mater. Chem.* **4**, 977 (1994).
2. Espeau, P., Reynolds, P. A., Bowling, T., Cookson, D. & White, J. W. X-Ray diffraction from layers of *n*-alkanes adsorbed on graphite. *J. Chem. Soc. – Faraday Trans.* **93**, 3201–3208 (1997).
3. McGonigal, G. C., Bernhardt, R. H. & Thomson, D. J. Imaging alkane layers at the liquid/graphite interface with the scanning tunneling microscope. *Appl. Phys. Lett.* **57**, 28–30 (1990).

4. Rabe, J. P. & Buchholz, S. Commensurability and Mobility in Two-Dimensional Molecular Patterns on Graphite. *Science* **253**, 424–427 (1991).
5. Venkataraman, B., Breen, J. J. & Flynn, G. W. Scanning Tunneling Microscopy Studies of Solvent Effects on the Adsorption and Mobility of Triacontane/Triacontanol Molecules Adsorbed on Graphite. *J. Phys. Chem.* **99**, 6608–6619 (1995).
6. Herwig, K. W., Matthies, B. & Taub, H. Solvent effects on the monolayer structure of long n-alkane molecules adsorbed on graphite. *Phys. Rev. Lett.* **75**, 3154–3157 (1995).
7. Endo, O. *et al.* Incommensurate crystalline phase of n-alkane monolayers on graphite (0001). *J. Phys. Chem. C* **115**, 5720–5725 (2011).
8. Ilan, B., Florio, G. M., Hybertsen, M. S., Berne, B. J. & Flynn, G. W. Scanning tunneling microscopy images of alkane derivatives on graphite: role of electronic effects. *Nano Lett.* **8**, 3160–3165 (2008).
9. Yang, T., Berber, S., Liu, J.-F., Miller, G. P. & Tománek, D. Self-assembly of long chain alkanes and their derivatives on graphite. *J. Chem. Phys.* **128**, 124709 (2008).

#6 (line 302) The flat-on layer does not describe the experimental data. So? A concluding sentence in the sense of ...this is why we suggest the layer to be formed from edge-on molecules? Why is that? Wouldn't one expect the opposite?

The orientation of the alkane molecules (edge-on, flat-on or mixed) and their exact distance is a very peculiar and hard to solve problem. From the early observations of the alkane monolayers on graphite there are arguments on both sides [1-2], still the majority of the (room-temperature) experiments found equidistant alkane-alkane distance and flat-on orientation. However, a temperature driven phase transition is reported by Dima *et al.*[3] and Endo *et al.*[4], where edge-on orientation together with mixed phases appear at low-temperature conditions. In our paper, we restricted ourselves to the examination of the two distinct, simple cases, the solely flat-on and the solely edge-on arrangements, which can be handled within the DFT methodology. As we have shown, the features of the observed moiré patterns in our samples are better reproduced by the simulated STM images of the edge-on orientation rather than the flat-on orientation at low temperature, in agreement with previous experiments [4]. However, we cannot exclude the possibility of mixed phases in the samples, as these geometries are beyond our DFT computational limit, thus no simulated STM images are available for the comparison. We have extended the manuscript to point out this limitation of our calculation and mention the possibility of mixed phase in our samples. Addition to the main text is as follows:

“However, from our DFT calculations we cannot rule out the presence of mixed phases in the molecule orientation, where the edge-on and flat-on configurations may be intermixed.”

1. McGonigal, G. C., Bernhardt, R. H. & Thomson, D. J. Imaging alkane layers at the liquid/graphite interface with the scanning tunneling microscope. *Appl. Phys. Lett.* **57**, 28–30 (1990).
2. Rabe, J. P. & Buchholz, S. Commensurability and Mobility in Two-Dimensional Molecular Patterns on Graphite. *Science* **253**, 424–427 (1991).
3. Dima, A. *et al.* Structure and phase transitions of monolayers of intermediate-length n-alkanes on graphite studied by neutron diffraction and molecular dynamics simulation. *J. Chem. Phys.* **131**, (2009).

4. Endo, O. *et al.* Incommensurate crystalline phase of n -alkane monolayers on graphite (0001). *J. Phys. Chem. C* **115**, 5720–5725 (2011).

Reviewer #2 (Remarks to the Author):

This paper describes that the identity of the common ambient contamination on the surface of vdW materials such as graphene, graphite, hBN and MoS₂ is composed of a self-organized molecular layer of normal alkanes with lengths of 20-26 carbon atoms, by using the low-temperature STM measurements and infrared spectroscopy. The relation between the contaminant layer and the anisotropic friction measured by AFM is also described. The main conclusion of the paper is well supported by the experimental results. Therefore, I would recommend this paper to be published after a minor revision concerning the following points.

1. Please specify the temperature of the IR measurement, because it is related to the phase of the specimen (smectic or crystalline).

This detail is clearly missing from the manuscript, we thank the reviewer for drawing our attention to this. The measurements were done at room temperature (~300 K) and the temperature is indeed a very important factor to understand the IR spectra and the structure of the samples.

Vapor-deposited, homogenous monolayers of C₂₄ or C₃₂ are in smectic phase at room temperature. The crystalline to smectic transition occurs between 215-230 K, while the smectic to liquid transition occurs around 340 K for the C₂₄ monolayers [1-3]. The respective values for the C₃₂ monolayer are just around 10 K above those temperatures. We cannot find an exact value for the C₃₂ monolayer's crystalline to smectic transition, while the smectic to liquid transition occurs at 350 K [1-2].

On the other hand, a slightly higher than monolayer coverage (1.15 monolayer), a denser package shifts these temperatures, resulting in a denser layer of C₂₄ that is still in a crystalline phase at room temperature, up to 310-320 K [1]. We suggest that the same applies for the denser C₃₂ layer as well.

We have extended the Methods section to clarify the measurement temperature.

1. Dima, A. *et al.* Structure and phase transitions of monolayers of intermediate-length n - alkanes on graphite studied by neutron diffraction and molecular dynamics simulation. *J. Chem. Phys.* **131**, (2009).
2. Hansen, F. Y., Herwig, K. W., Matthies, B. & Taub, H. Intramolecular and lattice melting in n-alkane monolayers: An analog of melting in lipid bilayers. *Phys. Rev. Lett.* **82**, 2362–2365 (1999).
3. Taub, H. *et al.* Slow Diffusive Motions in a Monolayer of Tetracosane Molecules Adsorbed on Graphite. in *AIP Conference Proceedings* vol. 708 201–204 (AIP, 2004).

2. As for the IR spectra shown in Figure 2, the splitting of the scissoring mode at 1472 cm⁻¹ is not attributed to the bilayer formation in ref. 40. On the contrary, it seems that the first layer in contact with the Ag(111) surface shows the split peaks. It may be attributed to the molecule-metal substrate interactions, which may not be the case for the HOPG surface. According to another paper of the same group (*J. Phys. Chem. B* 2000, **104, 7363-7369), split scissoring mode is attributed to the crystalline phase, not a double layer.**

We thank the reviewer for pointing out this clear mistake in our manuscript. Actually, we wanted to refer to *J. Phys. Chem. B* 2000, **104**, 7363–7369 [1]. In fact, the mistakenly referred

paper (Ref. 40 in the original manuscript, *J. Phys. Chem. B* 2000, **104**, 7370–7376) says only a little about the scissoring mode, it is discussing the effect of molecule-metal interaction mainly on the higher frequency stretching modes. Furthermore, the splitting of the scissoring mode closes as the layer thickness increases. We apologize for the mistake and thank the reviewer for the suggestion.

From the correct paper [1], to which the reviewer drew our attention, we understood that the splitting of the scissoring mode of the deposited alkane layer is attributed to an emerging crystalline phase. However, in the experiments presented by Yamamoto et al.[1] the peak splitting occurs only in thicker layers (>8.8 Å) of C44, which they described as a sign of the presence of more than two (1.: flat-on, 2.: gauche and 3+: crystalline) layers of C44. In our understanding, the splitting of the scissoring mode means a crystalline phase and a thicker layer.

In the same paper Yamamoto et al. [1] describes “the excluded volume effect” near the completion of the first monolayer, where in the denser layer the molecules tilt-out from the flat-on geometry and their molecular plane is perpendicular to the substrate, meaning a partly edge-on geometry. It has to be noted, that in this denser monolayer, no splitting of the scissoring mode was reported.

It could be possible that our vapor-deposited C32 layer is not an exact monolayer, but that it contains some areas with multilayers, consisting of crystalline phases. However, the fact that we could make stable STM imaging on the samples, which could be hardly possible in thick layer, imply that the regions consisting of thicker layers cover a small part of the surface.

We modified our interpretation on the C32 layer’s scissoring mode splitting in the light of the above-discussed two questions and added the correct reference. New text added to the manuscript:

“... the peak of the scissoring mode shows a splitting, which could be due to the presence of thicker layers of dotriacontane, with partial coverage, which could be in a crystalline phase³⁸.”

1. Yamamoto, M. *et al.* Structures of a Long-Chain n-Alkane, n-C44H90, on a Au(111) Surface: An Infrared Reflection Absorption Spectroscopic Study. *J. Phys. Chem. B* **104**, 7363–7369 (2000).

3. The STM contrast of normal alkane layer shown in Figure 3 seems to be well explained by the simulation. However, I wonder that the discrepancy between the distances obtained by the STM observation and DFT calculation in Figure 3 can be serious. The experimentally obtained intermolecular distances 0.42 nm and 0.57 nm are fairly larger than the simulated values of 0.378 nm, 0.485 nm, which is not a minor difference in my opinion. This difference makes the positional relation between the molecule and the substrate atoms different, and therefore, the real molecule-substrate interactions may not be reproduced by the simulation. Moreover, the larger intermolecular distance may allow a more dramatic orientation changes from the edge-on, beyond a slight tilt shown in Figure 3e. Hence, a more detailed simulation or discussion is required concerning this matter, such as the possibility of the further orientation changes. Even if the superlattice is not reproduced for the molecular arrangement with all the molecules with the flat-on orientation as shown in Fig. S11, it does not necessarily exclude the mixture of the edge-on, tilted-on, and flat-on orientation of the molecules.

We agree with the Referee that the difference of the intermolecular distance between the experiments and the calculations also makes the positional relation between the molecule and the substrate atoms different in the two cases. Since the molecules deposited from air are clearly not homogenous (see length distribution in Fig. 3c of the main text) we can't fully capture their behavior within DFT calculations, because of the need for periodic supercells. As we discuss in the main text, the following effects, observed in the STM are also reproduced in our DFT calculations:

1. Having two inter-molecule distances in the measurement.
2. The appearance of brighter and darker molecules in the STM images, due to the differing molecular environment. These brighter molecules form a quasiperiodic superlattice.

Both of these effects are captured by the DFT calculations of edge-on alkane molecules as a result of the differing molecular environments of the alkane chains on the graphite support. To further strengthen our observations, we have done calculations on much larger supercells with a larger inter-molecular distance (4.0 Å). These new calculations also reproduce the above two observations, as detailed below.

We added the following text to the manuscript:

“We have also checked that increasing the alkane – alkane distance to 4 Å in our calculation does not change the qualitative agreement with our STM measurement. This enlarged distance also results in a modulation of the apparent height of the molecules, corresponding to the measured quasiperiodic superlattice and the measured two inter-molecule distances of the alkanes (see Supplementary Figure 11b).”

Detailed discussion of the intermolecular distance:

In our calculations, we applied 3.8 Å distance for the edge-on orientation, as we have found this distance as the energetically most stable configuration in our reference DFT calculations, similarly to ref [1]. It is also known from previous DFT calculations [2] that the total energy of the structure depends weakly on the molecular distance between alkanes. Namely, for the edge-on orientation geometries have roughly the same energy on the graphite substrate in the range of 3.5-4 Å molecular distance. This allows us to increase the intermolecular distance in our calculations, since these geometries are close to equilibrium and can be also relevant in experiments. With the help of our new calculations, we have investigated the effect of the intermolecular distance on the Moiré patterns formed on the graphite substrate as the Referee suggested. It is worth noting that the increased molecular distance also significantly increases the commensurate supercell of the alkane-graphite system, making these calculations computationally challenging. More specifically, the Moiré wavelength has a divergence at 4.26 Å, increasing the needed supercell impossibly large to simulate even at 4.1 Å. In accordance with this, in our new DFT calculations we were able to reach only a maximal molecular distance of 4.0 Å before exceeding our computational limits. The supercell length then became 68.2 Å with 17 alkane chains. This geometry results in just over one thousand atoms (1159) in the simulation. The results of the two different molecule distances (3.8 Å and 4.0 Å) allows us to compare the main features of the observed Moiré periodicities and to make qualitative predictions.

The first important result of the new DFT calculation is the confirmation of the rotation of the alkane molecules, which is responsible for the geometrical effects in the observed Moiré patterns. In the case of the 4 Å molecular distance, we did not find evidence of more drastic rotation of the alkanes as was suggested by the Referee compared to the case of the previous

3.8 Å geometry. It is worth noting that our new calculation revealed a non-uniform rotation of the alkane molecules, namely that the molecules situated at a symmetric position over the graphite surface have much smaller rotation angle (see Figure below and FigS11b). Our results highlight the complex behavior of the rotation effect and show that the larger intermolecular distance does not result in increased rotation of the molecules.

As a next step, we have produced the simulated STM image for this geometry (see Figure below and FigS11b). The main important observation is that the simulated STM image is in qualitative agreement with our previous calculation. Namely, the fine details of the moiré pattern for increased intermolecular distances (4.0 Å) also show that additional periodicities appear below the supercell period in a similar way as for shorter molecular distances (3.8 Å). We observed the variation of the apparent height of the molecules along the stripe in the newly executed edge-on simulations, where the combined geometrical and electronic effects results in 4-5 brighter molecules together with 2-3 molecules with lower contrast. This variability in the simulated STM images, where different periodicities appear below the supercell period, is in good agreement both with our previous calculations and the experimentally observed STM images. It is worth noting that in contrast to the edge-on orientation, the flat-on geometry does not show such complex variability of the observed periodicities in our STM image simulations (see Fig S11c). This qualitative difference supports the rather edge-on nature of our findings.

We also quantified the observed periodicities by using the same FFT analysis. We found four peaks in the new geometry calculation at 0.4 nm, 0.52 nm, 1.7 nm and 3.4 nm values (see Figure). If we compare these values to our previous results (0.378 nm, 0.485 nm and 1.13 nm), where the molecule distance was 3.8 Å, we can recognize increased periodicities. This is not surprising for the first two values (0.378 nm, 0.485 nm), which correspond to geometrical effects (H atom positions). In those cases, the applied 4 Å molecular distance simply increases these periodicities, namely the molecular distance of the alkanes and the distance of the furthest hydrogens, respectively. The remaining two peaks (1.7 nm, 3.4 nm) also reflect electronic effects. This is in agreement with our previous findings, in which we revealed that the peaks with even larger values in the FFT spectrum originated from both geometrical and electronic structure effects. Our new calculation further confirms this observation, where periodicities with 1.7 nm and 3.4 nm values are obtained. These values are smaller than the supercell period and could be comparable with the experimentally observed 1D quasiperiodic patterns (of around 1.9 nm period). As we can see, the comparison of the two DFT calculations show a tendency in the observed periodicities, where the periodicities in the alkane-graphite system are shifting towards the larger values by increasing the molecular distance. Therefore, we can conclude that the distance between the alkane molecules changes the observed periodicities in the system, as the Referee mentioned. However, the main origins of these periodicities are the same as we have demonstrated by our additional DFT calculation. Therefore, we assume that further changes of the molecule distances (approaching the experimental values) will have the same qualitative behavior and will reproduce the observed periodicities of the measured STM topography images (0.42 nm, 0.57 nm and 1.95 nm).

Figure: Top: Relaxed geometry and calculated STM topographic image for 4.0 Ang alkane-alkane distance on a bilayer graphene surface. The calculation cell is 6.8 nm long. Gray lines with arrows show the two interchain distances and in red the quasiperiodic superlattice lengths. Bottom: Fourier transform of the simulated STM image.

Finally, we agree with the Referee that our calculations on pristine edge-on and flat-on geometries cannot exclude the possibility of mixed phases in our samples. Although the pure edge-on geometries show nice agreement with the experiments, mixed phases can also appear in samples. Unfortunately, we are not able to compare theoretical calculations of the mixed phase with the experiments, because these mixed phases are beyond our DFT computational limit. We have extended the manuscript to point out this limitation of our calculation and mention the possibility of mixed phase. Addition to the main text is as follows:

“However, from our DFT calculations we cannot rule out the presence of mixed phases in the molecule orientation, where the edge-on and flat-on configurations may be intermixed.”

1. Ilan, B., Florio, G. M., Hybertsen, M. S., Berne, B. J. & Flynn, G. W. Scanning tunneling microscopy images of alkane derivatives on graphite: role of electronic effects. *Nano Lett.* **8**, 3160–3165 (2008).
2. Yang, T., Berber, S., Liu, J.-F., Miller, G. P. & Tománek, D. Self-assembly of long chain alkanes and their derivatives on graphite. *J. Chem. Phys.* **128**, 124709 (2008).

Reviewer #3 (Remarks to the Author):

In the current paper, the authors have investigated the chemical composition of the airborne hydrocarbon contaminants on 2D materials using grazing angle IR, AFM, STM, ellipsometry and DFT calculation. They conclude that the contaminants are saturated alkane w/ 20-26 carbons that self-organize to form ordered "stripe" on the surface of 2D materials. The characterization and analysis are carefully conducted. The conclusion is supported by the experimental and DFT results. This is a high-quality work on a very important topic. While it has been recognized in the past decade that airborne hydrocarbon contamination significantly impacts the properties of 2D materials, it is still unclear what those contaminant are. Moreover, there has been very limited efforts on this topic. This reviewer suggest that the paper be accepted for publication after the following minor issues are addressed:

1. In grazing angle IR experiments, there should be another control: A hydrocarbon (e.g., SAMs) with functional groups (e.g., COO). This is to make sure the IR has the resolution to detect small amounts of functional groups.

We thank the reviewer for this suggestion, as the control experiments with hydrocarbons having functional groups, like carboxyl acids is truly necessary to further strengthen or disprove our statements. We have carried out additional IR measurements on samples with vapor deposited arachidic acid (C19 alkyl backbone + COOH carboxylic functional group) onto HOPG surfaces. We note, that we carefully reexamined the spectra presented in the original manuscript as well. We reevaluated our background subtraction process in a much more rigorous and reproducible way. We have added a description of the background fitting procedure to the Methods section:

“The reflectance spectrum was calculated by dividing the sample spectrum with the reference measurement. The extensive number of scans and the weak signal (0.05-0.1% change in reflection) necessitated the use of baseline correction. Due to the long-term instabilities of the interferometer, the instrumental artifacts result in broad spectral features⁶⁹. Artifacts are distinguishable from molecular vibrations which appear as narrow peaks in the spectrum. The baseline can be removed by manual baseline correction with proper attention^{69,70}. We used a cubic spline curve as baseline, which we defined by multiple points through the spectra with approximately 50-100 wavenumber step-size to correct the slowly changing background. Then, the absorbance was calculated as $A = 1 - R$ by using the corrected reflection (R) spectrum.”

Furthermore, we have made the raw data (STM, AFM and IR spectra) as well as the subtracted IR background available from Figshare at <https://doi.org/10.6084/m9.figshare.21294831>

We added a new figure (FigS13) and discussion to the Supplementary Information with the results of the arachidic acid control experiment.

In short:

1. We could detect the small amount of carboxylic functional groups in the IR spectrum of the evaporated monolayer of arachidic acid. The two most important spectral features of the arachidic acid monolayer are the appearance of C=O stretching mode between

1700 and 1800 cm^{-1} and the qualitatively different O-H or C-H stretching modes around 2800-3000 cm^{-1} .

2. As can be found in the literature carboxylic acids also self-organize on graphitic surfaces in a very similar way like alkanes, as they form lamellas (or stripes) and domains [1-2]. By PeakForce AFM we were able to image the stripes formed by the arachidic acid monolayer. See FigS13b. The self-organized arachidic acid monolayer also produced friction anisotropy domains owing to its long alkyl backbone (see FigS13c), we completed our explanation on friction domains with these results.

1. Rabe, J. P. & Buchholz, S. Commensurability and Mobility in Two-Dimensional Molecular Patterns on Graphite. *Science* **253**, 424–427 (1991).
2. Cyr, D. M., Venkataraman, B. & Flynn, G. W. STM Investigations of Organic Molecules Physisorbed at the Liquid–Solid Interface. *Chem. Mater.* **8**, 1600–1615 (1996).

2. While the authors discussed the alkane in the ambient (line 410-418), it is still unclear why it is the alkane with 20-26 carbons. Why not the smaller or larger alkanes?

The first monolayer of normal alkanes has an exceptionally high adsorption energy on graphite surface [1], which increases with their chain length [1-2]. At room temperature ($\sim 23^\circ\text{C}$) the shorter alkanes (<17 carbon atoms) do not form an ordered layer on the surface of graphite (desorbed or in liquid phase) [1,3,4], which gives a lower limit to the length of the spontaneously self-organized molecules from the ambient. On the other hand, the vapor pressure of the normal alkanes decreases significantly as their chain gets longer [5], which limits the number of longer molecules present in the ambient. However, owing to their higher vapor pressure at room temperature, there are more shorter alkanes than longer ones. One effect that could counteract this is the following: it was demonstrated that the longer alkanes can interchange shorter ones as they have higher adsorption energy [6].

As we are not acquainted with any particular process, which specifically favors the observed molecular lengths (20-26 carbon atoms), we can only speculate that the above-mentioned processes limit the length of the molecules building up the airborne layer.

1. Groszek, A. J. Selective adsorption at graphite/hydrocarbon interfaces. *Proc. R. Soc. London. A. Math. Phys. Sci.* **314**, 473–498 (1970).
2. Castro, M. A., Clarke, S. M., Inaba, A., Thomas, R. K. & Arnold, T. Preferential Adsorption from Binary Mixtures of Short Chain n -Alkanes; The Octane–Decane System. *J. Phys. Chem. B* **105**, 8577–8582 (2001).
3. Espeau, P., Reynolds, P. A., Bowling, T., Cookson, D. & White, J. W. X-Ray diffraction from layers of n-alkanes adsorbed on graphite. *J. Chem. Soc. - Faraday Trans.* **93**, 3201–3208 (1997).
4. Paserba, K. R. & Gellman, A. J. Effects of conformational isomerism on the desorption kinetics of n-alkanes from graphite. *J. Chem. Phys.* **115**, 6737–6751 (2001).
5. Chickos, J. S. & Hanshaw, W. Vapor pressures and vaporization enthalpies of the n-alkanes from C21 to C38 at T = 298.15 K by correlation gas chromatography. *J. Chem. Eng. Data* **49**, 620–630 (2004).
6. Xia, T. K. & Landman, U. Molecular Dynamics of Adsorption and Segregation from an Alkane Mixture. *Science* **261**, 1310–1312 (1993).

3. How local environment will impact the conclusion? In other words, how universal is the conclusion? The authors need to discuss on it.

Many groups from across the globe (Korea, China, USA, UK, Germany, ...) reported similar, yet unexplained stripe structures on the surface of different vdW materials, as we referred in the original manuscript Ref 6-17. Furthermore, other groups reported friction anisotropy (Ref. 12-14. and Ref. 18-23.) domains on vdW surfaces like graphene, graphite, hBN, MoS₂, WS₂, on the origin of which there was no consensus. We believe that our findings, namely to understand the chemical nature of the self-organized contaminant layer could explain a great majority of these observations. Our paper presents strong evidence that the friction domains are caused by self-assembled molecular lamellae, therefore we can conclude that there is a strong case to be made for alkanes as the source of the friction domains and stripe structures.

We further mention that mass spectroscopic measurements of the particulate pollution in ambient air in Sao Paulo and Salvador (Brazil) [1], as well as in Athens (Greece) [2] show a similar distribution of the alkane chain length as we observe in our STM measurements.

To make our case even stronger, we repeated the ambient deposition experiments in different geographic locations with freshly cleaved vdW crystals. We exfoliated graphite and hBN at Sardinia (Italy), San Diego (USA) and Szántód (countryside village in Hungary) and stored the freshly cleaved surfaces in ambient for 1-2 weeks. After bringing home the samples we immediately checked whether the self-organized monolayer formed or not. We think that the AFM experiments presented below act as solid evidence that the contamination layer forms also outside our local (Budapest) environment. We added the following text to the Results section of the main text to clarify this:

“The presence of these stripes is not restricted to our lab environment in Budapest. We have exposed exfoliated hBN and graphite crystals to the ambient air in San Diego (USA), Sardinia (Italy) and Szántód (Hungary). All of them show stripe patterns and friction anisotropy domains.”

a) Stripe pattern on the surface of bulk HOPG, 2 weeks after the cleavage in San Diego, USA
b) Stripe pattern on the surface of a thick (~100 nm) hBN flake, 1 weeks after exfoliation in Szántód, Hungary, c) A tripoint boundary of the friction anisotropy domains (shown by blue arrow) on an exfoliated HOPG flake, 1 week after exfoliation in Sardinia, Italy.

1. Caumo, S., Bruns, R. E. & Vasconcellos, P. C. Variation of the Distribution of Atmospheric n-Alkanes Emitted by Different Fuels' Combustion. *Atmosphere*. **11**, 643 (2020).

2. Andreou, G. & Rapsomanikis, S. Origins of n-alkanes, carbonyl compounds and molecular biomarkers in atmospheric fine and coarse particles of Athens, Greece. *Sci. Total Environ.* **407**, 5750–5760 (2009).

Reviewers' Comments:

Reviewer #1:

Remarks to the Author:

The authors have thoroughly addressed the issues raised. I recommend publication.

Reviewer #2:

Remarks to the Author:

My concern for the original version of the manuscript was fully addressed and I am satisfied with the authors' comment and revision. Therefore, I would recommend the paper should be published as it is.

Reviewer #3:

Remarks to the Author:

The authors have addressed the reviewers' comments and concerns. Now the paper is ready for publication.

Reviewer #1 (Remarks to the Author):

The authors have thoroughly addressed the issues raised. I recommend publication.

Reviewer #2 (Remarks to the Author):

My concern for the original version of the manuscript was fully addressed and I am satisfied with the authors' comment and revision. Therefore, I would recommend the paper should be published as it is.

Reviewer #3 (Remarks to the Author):

The authors have addressed the reviewers' comments and concerns. Now the paper is ready for publication.

We thank all reviewers for the insightful questions and comments, the responses to which have significantly improved our paper.